REGISTERED REPORT PROTOCOL

# Incarceration history and ethnic bias in hiring perceptions: An experimental test of intersectional bias & psychological mechanisms

Christopher R. Beasley[‡], Y. Jenny Xiao[‡]*

University of Washington Tacoma, Tacoma, Washington, United States of America

‡ CRB and YJX contributed equally to this work and co-first authors.
* beasley2@uw.edu

This is a Registered Report and may have an associated publication; please check the article page on the journal site for any related articles.

## Abstract

This study seeks to better understand mechanisms of bias against formerly incarcerated and ethnically minoritized job applicants as well as the interactive effects of those two identities. In a sample of 358 hiring managers in the United States, the 2 (incarceration history) x 4 (ethnicity) experiment will manipulate incarceration history and ethnicity through job application materials, and measure hireability, and perception of job applicants along dimensions of sociability/warmth, competence, and morality. We will use a moderated mediation model to test hypotheses regarding a main effect of prior incarceration and an interaction effect of incarceration history and ethnicity on judgments of hireability, as well as whether such effects are mediated through perception of job applicants. We expect results to inform both research and practice related to employment practices.

## Introduction

Mass incarceration is one of the most pressing social issues in the United States and results in collateral consequences that appear to have a disparate impact on people of color. The U.S. has a higher rate of incarceration than any other industrialized country in the world [1], with 3.11% of U.S. adults currently in prison, on parole, or formerly incarcerated [2]. Notably, the same study estimated that nearly one out of six Black men are currently in prison, on parole, or formerly incarcerated [2]. Although similar cumulative data are not available for other ethnicities, other reports have found that Latinx, Native American, Pacific Islander, American Indian, and Alaskan Native individuals are incarcerated at much higher rates than their White counterparts and the general public [3, 4]. Individuals with incarceration history appear to experience interrelated collateral consequences such as unemployment, which was 27.3% for formerly incarcerated people in 2008 compared to 5.2% for the general population [5]. Such unemployment trends appear much more likely among formerly incarcerated Hispanic and Black men than White men. For example, the unemployment rate for formerly incarcerated people in 2008 was 18.4% for White men, 26.5% for Hispanic men, and 35.2% for Black men

**Data Availability Statement:** All research design, materials, hypotheses, and planned analyses will be pre-registered prior to data collection. In accordance with open science practices, all measures, experimental conditions, data exclusions, and methods of determining sample size are reported below. Anonymized data will be made available at the time of publication in on online public repository (e.g., https://osf.io/).

**Funding:** Funded by University of Washington Tacoma School of Interdisciplinary Arts and Sciences The funders had and will not have a role in study design, data collection and analysis, decision to publish, or preparation of the manuscript.

**Competing interests:** The authors have declared that no competing interests exist.

[5]. Although there has been some (still limited) research on the main and interactive effects of criminal history and ethnicity on biased hiring outcomes, little is known about the psychological mechanisms through which these biases may occur. Therefore, the current study seeks to better understand the psychological mechanisms for the unemployment collateral consequence.

## Incarceration history and hireability

One likely reason for considerably higher unemployment rates among formerly incarcerated people is their criminal records. Employer screenings of such records have become increasingly widespread. Batastini and colleagues [6] found a causal link between criminal history and a host of outcomes in attitudes and intentions among undergraduate participants, including perceptions of acceptability of job applicants, the extent to which the applicant's current problems were their fault, how dangerous the applicant was, preferred social distance with the applicant, and likelihood to help. Batastini and colleagues [7] replicated this study with Human Resources (HR) professionals and found similar causal effects of criminal history on judgements of applicants. Similarly, in a recent nationwide survey, [8] manipulated incarceration history and job applicant's ethnicity (Black or White) and measured participants' perception of job applicants. These results confirmed that participants were apprehensive about applicants with incarceration history, which was consistent with much of the existing work, providing converging evidence for the unemployment consequences for individuals with incarceration history. In the current research, we will first examine the effect of incarceration history on judgment of hireability, in our first hypothesis.

$H_1$: *Incarceration history will have a direct effect on judgment of hireability.*

## Racial inequities in hireability

The detrimental effects of incarceration history for employment can be exacerbated for people of color. Numerous studies have shown a causal link between ethnicity and employment decisions, and a recent meta-analysis found this effect does not appear to be diminishing [9]. These authors found that White applicants received 36% more callbacks than Black applicants and 24% more than Latinx applicants.

Importantly, such racial bias may intersect with criminal history, although the precise effects of such an interaction seems inconclusive given the existing empirical evidence. In an experiment where applicants' ethnicity and criminal history (none vs. drug-related felony) were manipulated, 34% of White applicants without criminal records received call-backs followed by White applicants with criminal records (17%), Black applicants without criminal records (14%), and Black applicants with criminal records (4%; [10]). Similar interactions have been found for Latinx applicants when compared to White applicants. For example, in an experiment, researchers found main effects of education, work experience, and criminal history (none vs. drug-related misdemeanor vs. drug-related felony), but no main effect of ethnicity (Latinx vs. White), on dichotomous hiring decisions and strength of recommendation [11]. Although there was an interaction between ethnicity and criminal history affecting strength of recommendation, the interaction didn't emerge for hiring decisions. Importantly, Latinx applicants with felony drug convictions were rated lower on strength of recommendation than Latinx applicants with no criminal history; whereas, White applicants were rated similarly regardless of conviction history. In a similar study, the same researchers found main effects for job qualifications and type of criminal history on both hiring decision and strength of

recommendation, but no effect of ethnicity (Latinx vs. White) nor any interactions between ethnicity and other factors [12].

Furthermore, Reicher [13] examined employment outcomes for job applicants with incarceration history in Australia, by experimentally manipulating job candidates' ethnicity (Caucasian Australian vs. Indigenous Australian). Employers expressed somewhat lower willingness to hire an Indigenous Australian job candidate, compared to their Caucasian Australian counterparts, but this finding was only marginally significant. Most recently, DeWitt and Denver [8] found that while job applicants with an incarceration history were viewed more negatively in the hiring context, ethnicity (Black or White) did not moderate such effect. Importantly, DeWitt and Denver [8] posited several possible reasons for the null effect of ethnicity, such as their survey methodology, and the possibility that certain social categories could dominate our perception when encountering multiple intersectional social categories.

Collectively, these findings suggest a relatively consistent effect of incarceration history on hireability, but provide rather mixed evidence on the interaction between incarceration history and ethnicity. This could be, at least partially, due to the inconsistency of the specific ethnic and/or racial groups included in existing research, inclusion of different additional variables (e.g., education, job qualification, type of crime), as well as the idiosyncratic choices of professions or jobs. In the current research, we examine how incarceration history interacts with ethnicity by including four ethnicities—White, Black, Asian, Latinx—to affect judgments of hireability, and several possible psychological mechanisms through which such effects could occur.

H₂: *The interaction between incarceration history and ethnicity will have a direct effect on hireability ratings*

## Social categorization

One potential explanation for the main effects of criminal history and ethnicity on hiring perceptions may be self and social categorization. Self and social categorization entail grouping people in a manner that makes sense to the perceiver and structures our social environment [14]. We may categorize ourselves based on an individual identity, a collective identity, or both, according to the immediate social and motivational context–a process known as self-categorization [15]. Indeed, there is extensive evidence that self-categorization with a social group can influence perception of the social environment, leading to biases in memory [16], evaluation [17], and behavior [18]. These intergroup biases could ultimately hurt employment opportunities for individuals with incarceration history, especially when the majority of people making hiring decisions do not have incarceration histories themselves.

Similar to the self-categorization process, we approach others in social situations through a person-based processing mode of impression formation, or a category-based processing mode, which could affect everything from attention to behavior towards the individual [19]. Whereas person-based processing focuses on the representation of the unique individual, category-based processing revolves around a category prototype and its associated group stereotypes [19]. It has been theorized and empirically shown that affective, evaluative, and behavioral biases occur more likely when category-based judgment processes dominate impression formation, since attitudes and behaviors towards group members are derived more from attitudes towards the group as a whole rather than direct experiences with particular individuals (e.g., [19, 20]).

Recently, Young and Powell [21] proposed a theoretical model explaining mechanisms through which impression formation could produce such bias. This model emphasized the role of hiring managers' perceptions of warmth and competence of ex-offender job applicants in their hiring decisions. Young and Powell [21] argued that the lack of interest and/or motivation to gain additional individuating information about ex-offender job applicants may result in largely category-based processing of such applicants. Therefore, it is possible that the largely category-based processing of ex-offender job applicants elicits negative stereotypes of this group, placing them at a disadvantage during the hiring process.

## Intersectionality

Additional social psychological concepts and theory suggest schemas may be a psychological mechanism through which ethnicity may interact with other variables to produce disparate outcomes. As types of schemas [22], stereotypes are complex, with subtypes and subgroups of superordinate stereotypes developing [23]. Considering the intersectionality of multiple social categories and identities (e.g., gender and ethnicity) allow us to predict and understand experiences of unique forms of prejudice and discrimination, distinct from only considering a single aspect of one's identity (e.g., ethnicity) [24]. Such intersectional effects have been demonstrated for ethnicity and gender in stereotyping [25] and implicit bias [26], as well as for ethnicity and height in threat perception and police stops [27].

According to the framework of gendered race theory, racial stereotypes contain a gendered component whereby certain racial and ethnic groups are stereotyped as more masculine or feminine (e.g., [28]). The gendered race phenomenon has been shown to have important implications not only for categorization and perception of individuals [29], but also for stereotype content. For instance, Asian men and Black women are viewed as less prototypical of their race categories [28]. Such gendered racial stereotypes have been shown to harbor implications for interracial marriages, leadership selection, and athletic participation [30].

It is also possible, as DeWitt and Denver [8] pointed out in their writing, that certain social categories could dominate our perception when encountering multiple intersectional social categories. For instance, Rattan and colleagues [31] found that evaluation of job candidates depended on which social category was more salient in the given context, even when the job candidates were perceived to belong to multiple social categories. In a stereotypically male and stereotypically Asian employment context, participants rated an Asian American female applicant as more hirable when her ethnicity, rather than gender, was made salient [31].

To date, little to no theoretical or empirical work exists regarding intersectionality between ethnicity and incarceration history. However, intersectional stereotypes and the gendered race theoretical perspective provide a general framework for making predictions that ethnicity and incarceration history can combine to create unique subgroups of incarceration stereotypes for people of various ethnicities.

## Fundamental dimensions of impression formation

As noted earlier, warmth and competence appear to be important factors in judgements of people with criminal records (e.g., [21]). Warmth and competence have been established, both theoretically and empirically, as two fundamental dimensions along which we form impressions of individuals as well as social groups (e.g., [32]). According to this framework, while warmth judgments allow us to understand others' intentions including trustworthiness, competence judgments allow us to evaluate the ability of others. Importantly, warmth and competence judgments are not only central to perception of individuals, they also underlie group

stereotypes. The warmth and competence dimensions of specific group stereotypes have been shown to predict distinct patterns of responses to members of these groups [33].

Although ex-offenders are not a group originally mapped onto the competence/warmth dimensions by [21, 33], argued that perception of ex-offenders may be synonymous with populations such as those perceived to be poor or receiving public assistance (who are mapped in the low competence/low warmth quadrant; [33]). Moreover, because White people have been shown to elicit perceptions of high competence/high warmth reactions (compared to racial minority groups), this model [21] also proposed that racial and ethnic minority ex-offenders would be perceived to be less warm and less competent compared to their White counterparts [21]. Because this model regarding the central role of hiring managers' perceptions of warmth and competence of ex-offender job applicants in hiring decisions was purely theoretical, it is important to empirically evaluate its tenets.

More recently, a newly emerging perspective and empirical evidence points to moral character as a third independent dimension of impression formation of people and groups (e.g., [34–37]). Importantly, such a three-dimensional model is theorized to apply not just to perception and judgment of individuals, but that of social groups, such as social stereotypes [34]. Providing strong support for the addition of moral character as a unique third dimension, Brambilla and colleagues [36] showed that evaluations of an unfamiliar ethnic group were influenced more by ratings of the group's moral traits than by ratings of its sociability/warmth or competence traits.

A prevalent perception of a criminal record is that it signals that a person is untrustworthy [38, 39]. In recent research [40], Mikkelson and Schweitzer manipulated incarceration status, length of incarceration, time since release from prison, and measured perception of morality and hiring decisions. As a result, perceived morality of the applicant mediated the relationship between incarceration status and hiring decisions, such that previously incarcerated applicants were perceived to be less moral, leading to lower likelihood of being hired [40]. Given this collection of previous research and theorizing, in the current work we explore whether perceptions of competence, warmth, and moral character could explain, at least partially, the intention to hire individuals with incarceration history (compared to their counterparts without) (Fig 1).

$H_3$: *The effect of the interaction between incarceration history and ethnicity on hireability ratings will be mediated by judgments of morality, sociability/warmth), and competence*

## Methods

All research design, materials, hypotheses, and planned analyses will be pre-registered prior to data collection. In accordance with open science practices, all measures, experimental conditions, data exclusions, and methods of determining sample size are reported below. Anonymized data will be made available at the time of publication in on online public repository (e.g., https://osf.io/).

### Participants

We will recruit an online sample of workplace supervisors in the United States. G*Power 3.1 suggests collecting from 358 participants to achieve power of .80 at alpha of .05 for a linear multiple regression type model to detect small-to-medium effects ($f^2$ = .05) with the number of predictors included in the model. We will recruit about 10% more in order to meet the target sample size of 358 after attrition and/or exclusion of data (e.g., based on manipulation checks).

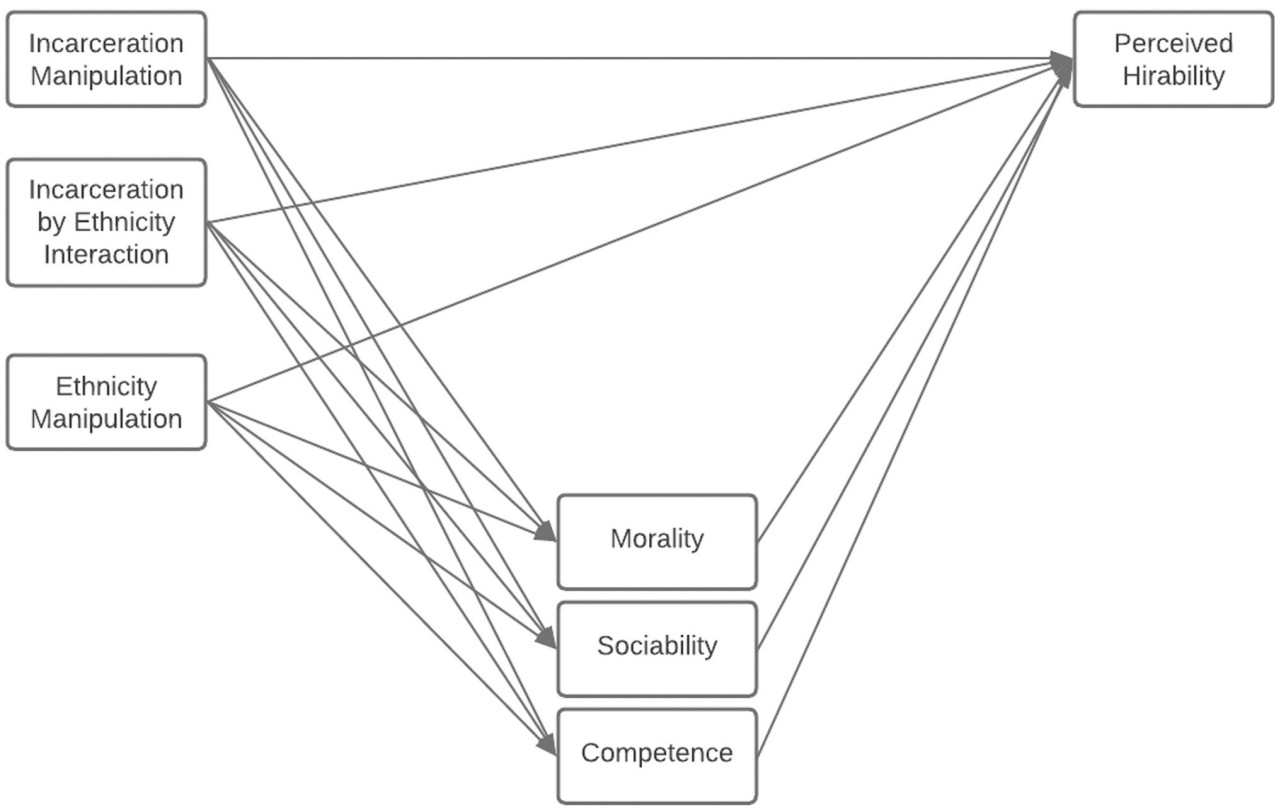

**Fig 1. Conceptual model for effect of incarceration history and ethnicity on perceived hireability.**

We will contract with CloudResearch to manage recruitment through Amazon Mechanical Turk using their CloudResearch Toolkit. We will establish quotas for this sample to better ensure representation of people involved in hiring decisions. Such quotas will approximate 74.9% women, 61.3% White Non-Hispanic, 11.9% Black, 6.16% Asian, 2.01% Mixed Ethnicity, 0.53% Native American, 18% Native Hawaiian or Pacific Islander, and an average age of 45 years. Our data collection will require that potential participants have an approval rating of 95% or more, and prospective participants whose past responses CloudResearch has designated as low-quality will be ineligible for the study. Each participant will be compensated $6 for the 30-min online experiment. We have obtained approval from the University of Washington Human Subjects Division (IRB) for all procedures in this research.

## Design

This study will employ a 2 (incarceration history) x 4 (ethnicity: Asian, Black, Latinx, White) between-subjects randomized experimental design, with judgments of hireability as the dependent variable. We will control applicant gender, qualifications, and other characteristics by holding them constant across all conditions.

## Materials

**Participant instructions.** We will provide participants with an overview of the study (see S1 Appendix) that describes the context of a simulated hiring process for a Human Resources Manager position and an overview of the materials they will be reviewing. We selected this

position to enhance generalizability of past research to white-collar hiring. Participants will be instructed to approach the hiring decision-making process as they would in their job and to evaluate the applicant to the best of their ability even when information may be limited.

**Job materials.**   We will manipulate job applicant ethnicity [Asian, Black, Latinx, or White] and prior incarceration history [previously incarcerated or not previously incarcerated] through the job application materials and highlights from a hiring assistant's job interview with the applicant. The job materials [see Appendices B and C] will include an advertisement for the position, the candidate's cover letter and resume, and highlights from a hiring assistant's interview with the applicant. The job advertisement will be for a Human Resources Manager position and will include information about the position, its responsibilities and roles, and required as well as preferred qualifications. We adapted a job advertisement for a Human Resources Manager position at the authors' university for use in this study to maximize ecological validity. The same job advertisement is used in all conditions.

The cover letter, resume, and interview will all include information about the candidate's involvement in a staff association. We will manipulate the ethnicity of the applicant in these materials by using a stereotypical name (e.g., Kevin Ming Lee as a stereotypical Asian name), and through their staff association experiences (e.g., Chinese American Association). We use names that have been shown to be overwhelmingly perceived to signal each ethnicity (e.g., [41, 42]). For instance, in a previous study, the Black name Jamal was perceived as Black among more than 95% of respondents [42]. In a separate study, it was shown that when Hispanic first names were paired with Hispanic last names (as opposed to White last names), the individual was perceived as Hispanic more than 90% of the time [42]. However, to offset potential covariance of minoritized names with lower SES, we used an atypical White name of Kody, because atypical names are more likely to be given by mothers with lesser education [43]. We will manipulate incarceration history in these materials by including terms related to previous incarceration in their staff association experiences (e.g., Formerly Incarcerated Future Professionals Network) or not. The interview summary will also include information about a challenge the applicant overcame, which will speak to either their previous incarceration (incarceration history) or a different challenge in life (no incarceration history). We will further manipulate ethnicity and incarceration history by discussing challenges related to the candidate's ethnicity and reintegration after prison (see interviews in S3 Appendix).

**Person perception measures (morality, sociability/warmth, competence).**   After viewing the application, participants will be asked to indicate on a 7-point scale, anchored by 1 (*not at all*) and 7 (*very much*), the extent to which each of the nine traits characterizes the job candidate they just read about (S4 Appendix). The measure is adapted from [44], with three morality traits (i.e., sincere, honest, trustworthy), three sociability/warmth traits (i.e., friendly, warm, likeable), and three competence traits (i.e., intelligent, competent, skillful). Traits will be presented in randomized order.

**Perceived hireability.**   We will use the Hireability Scale [45] to assess the extent to which participants perceive applicants as hireable. This is a 4-question, 9-point Likert-type (1 = Not at All to 9 = Very Much) self-report instrument (S5 Appendix) with prompts such as, "How likely would you be willing to hire this candidate?" and "To what extent is this a top-notch candidate?" The scale was found to have high internal consistency (Cronbach's alpha = .99) and high predictive validity, correlating with job candidate characteristics, in previous research [45].

**Manipulation checks.**   We will assess the manipulation effectiveness by asking three multiple choice questions subsequent to measurement of the dependent variables (see S6 Appendix). The first question, "What ethnicity was the job applicant?," will include four response options—"Asian", "Black", "Mexican", "White". The second question, "Has the job applicant been to prison or otherwise incarcerated?," will include two response options—"No, the

applicant has not been to prison or otherwise incarcerated" and "Yes, the applicant has been to prison or otherwise incarcerated." Lastly, if participants indicate the applicant has been incarcerated, we will ask whether that incarceration occurred as a result of a blue or white collar crime. This exploratory question is meant to provide us with information regarding participants' perception of type of crime, without us providing this information explicitly.

**Demographic questionnaire.** The final instrument participants will complete is the Demographic Questionnaire (see S7 Appendix). This includes open-ended and forced choice questions about the participant's traits and characteristics. The open-ended questions will inquire about participants' age, gender, ethnicity, and occupation, and personal experience of incarceration. The closed-ended questions will inquire about their annual income bracket, level of current and completed education, incarceration history, existing close relationships with incarcerated people.

## Procedures

**Recruitment.** We will be using CloudResearch's MTurk Toolkit to gather data through the Amazon MTurk online research platform. CloudResearch will be contracted to manage this recruitment, with our research team providing specifications for the sample. As noted in the sample description, we will establish quotas to ensure greater representation of the Human Resources Management field.

**Informed consent.** The link to our study directs prospective participants to an informed consent page on Qualtrics. This informed consent page will include an overview of the study, the purpose of the study, procedures participants can expect, risks involved with participation, benefits of participation, assurances of anonymity, a summary of how data will be used, and who to contact if participants have concerns or harms to report. We will include multiple choice questions about the study to help ensure comprehension of the information presented. Prospective participants will then sign the informed consent form digitally. Participants will not be able to advance past this page until five minutes have elapsed and they have answered each of these questions.

**Job ad.** Qualtrics will then direct consenting participants to the job ad for a Human Resources Manager position (see S2 Appendix). The job ad page will also include a set of questions about the position to help draw the participant's attention to this information. These questions assess the participant's recall about the position, its responsibilities and roles, and both required and preferred qualifications (see S2 Appendix). Participants will not be able to advance past this page until five minutes have elapsed and they have answered each of these questions.

**Manipulation.** Following informed consent, Qualtrics will randomly direct participants to the job application materials for one of the eight experimental conditions, (1) formerly incarcerated Asian applicant, (2) formerly incarcerated Black applicant, (3) formerly incarcerated Latinx applicant, (4) formerly incarcerated White applicant, (5) Asian applicant with no indication of incarceration history, (6) Black applicant with no indication of incarceration history, (7) Latinx applicant with no indication of incarceration history, White applicant with no indication of incarceration history. The application package (see S3 Appendix) includes a cover letter, a resume, and highlights from a hire assistant's interview with the applicant. In addition to these job materials, the manipulation includes a set of questions about the applicant to help draw the participant's attention to this information. These questions will assess participants' recall about the candidate's name, their level of education, the candidate's years of experience in the role, the experiences the candidate has had, the candidate's volunteer experiences, and why the candidate is interested in the position (see S3 Appendix).

Participants will not be able to advance past these pages until five minutes have elapsed and they have answered each of these questions.

**Data collection.** Following the manipulation, Qualtrics will direct participants randomly to the dependent variable measures in randomized order—morality, sociability/warmth and competence, and hireability. After participants have completed all of the dependent variable measures, Qualtrics will direct them to a page with the three manipulation check questions (see S6 Appendix). Qualtrics will then direct participants to a page with demographic questions (see S7 Appendix). Lastly, Qualtrics will present those who complete the study with a final confirmation page that acknowledges their contributions to the research.

## Planned analyses

We will first exclude data from participants who fail our attention and/or manipulation check from data analysis. It is hypothesized that a moderated-mediation relationship will exist whereby manipulating the incarceration history of the job applicant (X) will decrease participants' perceived hireability (Y) especially for racial minority applicants (W–moderator) via multiple parallel mediators (Ms): competence, sociability/warmth, morality. All continuous variables will be mean centered, and nominal predictors will be dummy coded. We will use the PROCESS moderated mediation Model 8 [46], with Y = hireability rating; X = incarceration history; parallel Ms = competence, sociability/warmth, morality; W = ethnicity of job applicant (White, Black, Asian, Latinx).

## Expected findings

Conditional direct and indirect effects will be examined using PROCESS Model 8 [46] with incarceration history as the primary independent variable (x); ethnicity of job applicant (White, Black, Asian, Latinx) as the moderator (w); competence, sociability/warmth, and morality as parallel mediators; and hireability as the dependent (y). All indirect effects will be computed for each of 10,000 bootstrapped samples and will be considered significant if their 95% confidence intervals do not include zero, as these bootstrapped effects do not produce exact *p* values.

We expect that the overall model would predict a significant percentage of the variance in hireability ratings. Of the individual predictors, if Hypothesis #1 and #2 are supported, we expect to see that incarceration history and the incarceration history x applicant ethnicity interaction will all predict significant unique variance in hireability ratings. Specifically, we expect that applicants with incarceration history will receive lower hireability ratings.

If our Hypotheses #3 is supported, we also expect that the incarceration history x applicant ethnicity interaction would have significant indirect effects on hireability ratings through all mediators (competence, sociability/warmth, and morality). We expect that increased feelings of competence, sociability/warmth, and morality should all predict higher levels of hireability ratings.

## Discussion

This study addresses a pressing social issue in the United States—the collateral consequences of unemployment after incarceration. Such unemployment is estimated to be five times that of people who have not been incarcerated [5]. The current study seeks to better understand the mechanisms for the unemployment collateral consequence and its disparate impact on people of color. Prior inquiry into this topic suggests that job applicants with criminal records are viewed more negatively than those without such records [6], and that this effect may be more pronounced for non-White applicants [10]. Previous research further suggests some potential mechanisms for the hireability bias such as perceptions of formerly incarcerated applicants as

less moral (e.g., [40]). However, research examining the psychological mechanisms underlying the hireability bias for formerly incarcerated individuals are still scant, although we have outlined theoretical support for other mechanisms such as warmth and competence judgments [21]. Given this body of literature, we developed the following four hypotheses.

$H_1$: *Incarceration history will have a direct effect on perception of hireability.*

$H_2$: *The interaction between incarceration history and ethnicity will have a direct effect on hireability ratings.*

$H_3$: *The effect of the interaction between incarceration history and ethnicity on hireability ratings will be partially mediated by judgments of morality, sociability/warmth, and competence*

To test these hypotheses, we will be conducting an online 2 (incarceration history) x 4 (ethnicity: Asian, Black, Latinx, White) between-subjects randomized posttest-only experiment with 358 workplace supervisors in the United States. Recruitment and data collection will be managed by CloudResearch through Their Toolkit for Amazon Mechanical Turk. We will present participants with a fictitious job ad as well as accompanying materials from a fictitious applicant, with the applicants' incarceration history and ethnicity manipulated through stereotypical names, organizational affiliations, and explanations of gaps in work history. We will then assess our dependent variables of perceived hireability as well as potential mediators such as morality, sociability/warmth, and competence. We will analyze a moderated mediation model to examine whether incarceration history has an effect on perceived hireability, whether there is an interaction between incarceration history and ethnicity, and whether morality, sociability/warmth, and competence partially mediate this effect. We expect to find a main effect for incarceration history, and interaction with ethnicity, and partial mediation through the aforementioned factors.

The limitations of our design primarily center on external validity. Although our position is one of a Human Resource Manager, our participants will likely hold a variety of management roles rather than the more specific HR Director role that would typically oversee hiring of this position. However, given our limited access to HR Directors, we are not able to circumvent this limitation. Similarly, we are not randomly sampling from the complete population of HR Directors, because we do not have access to a complete database of said directors. Given the roles of our participants and the lack of random sampling, our results are not likely to fully generalize to the population of HR Directors. Additionally, our manipulation and participants' perceptions of candidates are occurring through a simulated situation. Specifically, we recognize that real-life hiring situations often involve interactions with the candidates (e.g., interviews), comparing certain candidates with others applying for the same position, and may come with more consequential decision making processes. Such features of our simulated hiring setting may limit the generalizability of our findings. However, despite such limitations, the current experimental design offers us the opportunity to closely control the manipulations and allows us to make stronger arguments regarding the causal nature of the relationships we are examining in our model. Similarly, we expect random assignment to conditions will control extraneous participant variables such as participant attitudes and beliefs, prior exposure to formerly incarcerated people, exposure to people of different ethnicities, and tendency to respond in socially desirable ways.

## Significance and conclusions

Despite these limitations, we expect findings to advance the field's understanding about the effect of incarceration history and ethnicity on the hiring process. In particular, we expect

this research will foster a greater understanding about psychological mechanisms for biases in such hiring. Although some empirical evidence of bias exists, currently such psychological mechanisms are not well understood. We believe a better understanding of such mechanisms will facilitate the development of interventions to prevent and/or reduce such bias. However, further research may be needed to better understand the ways in which people develop these perceptions and ways to alter them. We also expect that future research will be able to use these findings to support investigations of other mechanisms as well as whether such mechanisms are dependent on the types of crimes people have been convicted of.

## Supporting information

**S1 Appendix. Participant instructions.**
(DOCX)

**S2 Appendix. Job ad.**
(DOCX)

**S3 Appendix. Job applicant materials.**
(DOCX)

**S4 Appendix. Measures of morality, sociability/warmth, and competence.**
(DOCX)

**S5 Appendix. Measure of perceived hireability and hiring decision.**
(DOCX)

**S6 Appendix. Manipulation check questions.**
(DOCX)

**S7 Appendix. Demographic survey.**
(DOCX)

## Author Contributions

**Conceptualization:** Christopher R. Beasley, Y. Jenny Xiao.

**Data curation:** Christopher R. Beasley, Y. Jenny Xiao.

**Formal analysis:** Christopher R. Beasley, Y. Jenny Xiao.

**Funding acquisition:** Christopher R. Beasley, Y. Jenny Xiao.

**Investigation:** Christopher R. Beasley, Y. Jenny Xiao.

**Methodology:** Christopher R. Beasley, Y. Jenny Xiao.

**Project administration:** Christopher R. Beasley, Y. Jenny Xiao.

**Resources:** Christopher R. Beasley, Y. Jenny Xiao.

**Software:** Christopher R. Beasley, Y. Jenny Xiao.

**Supervision:** Christopher R. Beasley, Y. Jenny Xiao.

**Validation:** Christopher R. Beasley, Y. Jenny Xiao.

**Visualization:** Christopher R. Beasley, Y. Jenny Xiao.

**Writing – original draft:** Christopher R. Beasley, Y. Jenny Xiao.

**Writing – review & editing:** Christopher R. Beasley, Y. Jenny Xiao.

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
