## [Decision Letter · Decision Letter 0]

2 May 2022

PONE-D-22-02329Incarceration History and Ethnic Bias in Hiring Perceptions: An Experimental Test of Intersectional Bias & Psychological MechanismsPLOS ONE

Dear Dr. Beasley,

Thank you for submitting your manuscript to PLOS ONE. After careful consideration, we feel that it has merit but does not fully meet PLOS ONE’s publication criteria as it currently stands. Therefore, we invite you to submit a revised version of the manuscript that addresses the points raised during the review process. You should be pleased to note how enthusiastic the reviewers and I were to read this protocol. Revision to include the suggestions from the reviewers will strengthen the protocol as written and also prepare the study team for eventual publication of the study findings.

We look forward to receiving your revised manuscript.

Kind regards,

Andrea Knittel

Academic Editor

PLOS ONE

Journal Requirements:

Reviewers' comments:

Reviewer's Responses to Questions

**Comments to the Author**

1. Does the manuscript provide a valid rationale for the proposed study, with clearly identified and justified research questions?

Reviewer #1: Partly

Reviewer #2: Yes

2. Is the protocol technically sound and planned in a manner that will lead to a meaningful outcome and allow testing the stated hypotheses?

Reviewer #1: Yes

Reviewer #2: Partly

3. Is the methodology feasible and described in sufficient detail to allow the work to be replicable?

Reviewer #1: Yes

Reviewer #2: Yes

4. Have the authors described where all data underlying the findings will be made available when the study is complete?

Reviewer #1: No

Reviewer #2: No

5. Is the manuscript presented in an intelligible fashion and written in standard English?

Reviewer #1: Yes

Reviewer #2: Yes

6. Review Comments to the Author

You may also provide optional suggestions and comments to authors that they might find helpful in planning their study.

Reviewer #1: Overall Comments:

- The authors have chosen a timely and important topic of focus. I suggest that the purpose of the paper be more explicitly aligned with the model that is presented. Further, the paper is structures in a clear and articulate way however, there is limited literature for each of the categories and I suggest the authors expand here. The literature requires to be woven together further rather than step by step overview of a few selected studied per category. The study generally requires further rooting in the existing literature. There are grammatical errors through, the authors are encouraged to carefully edit the document. It is noted that the data will be made available, however it is not clear where.

Review:

- need further context as it pertains to employment outcomes and experiences for formerly incarcerated persons (i.e., exploitation in employment, underemployment, industries that do hire vs those that don’t, etc.)

- the authors mention that the intention of the study is to “better the mechanisms r the unemployment collateral consequence understand and its disparate impact on people of color”; if the outcomes were related to impact on people of colour then how come the focus is on hiring manager decision making? If this was the case, wouldn’t formerly incarcerated people of colour be the population and sample of focus for the study? Instead, this study seems to point to employer decision making and perhaps what factors might impact their hiring decisions, that in turn may better help to explain employment outcomes for persons, instead, impact is suggestive of an internal effect and would require the perspectives of the persons of interest.

- The authors note that “employer screenings of such [criminal records] have become increasingly widespread”, there are other contributing factors – availability of record through media and internet search, and especially in the U.S. seeking information through various sources – this context is missing.

- The literature supporting hypothesis 1 is quite limited. For example how about scholarship which points to employer willingness to hire (e.g. SHRM & Chalres Koch Institure, 2018)

- Each study is described separately in the literature review, I suggest writing this cohesively and demonstrating how the results and findings from each relates to the other in some way.

- limited literature referenced for intersectionality. I would like to see more here. This is relevant but the leap from how this is currently conceptualized to how it will be approached with this population is much too big. The authors need to close this gap, perhaps with further literature and/or theorizing.

- Limited explanation of the warm and competence model, expand

- The authors cite “Goodwin et al., 2041” – please correct this with the correct citation

- Warmth, Morality, sociability, competence are not clearly delineated as distinct constructs. Sociability is not clearly conceptualized.

- Under participants, the authors mention that they will recruit? The language throughout this section references what the authors will do? Is this what was done? Rephrasing may be required.

- Is Latinx synonymous with Chicanx? The terms seem to be used interchangeably. Please explain or use one term for consistency.

- Is there a reason for focusing on human resource managers when the outcome is hiring? How about expanding to hiring managers in general?

Reviewer #2: The authors are proposing to carry out a much-needed study about the mechanisms underlying discriminatory processes in hiring by race/ethnicity and incarceration history. We know stark disparities exist in employment/hireability by both of these characteristics, but we know much less about why. My comments are meant to strengthen the proposed design and connect these authors to criminological and sociological work that might bolster the contribution of this study. My comments are as follows: (see attachment)

7. PLOS authors have the option to publish the peer review history of their article (what does this mean?). If published, this will include your full peer review and any attached files.

Reviewer #1: No

Reviewer #2: No

---

## [Author Response · Author response to Decision Letter 0]

25 Oct 2022

COMMENTS BELOW ARE ALSO UPLOADED AS A FORMATTED DOCUMENT 

Dear Editor Andrea Knittel,

We are grateful for the opportunity to revise and resubmit our manuscript “Incarceration Hias & Psychological Mechanism” to PLOS ONE. We were pleased to hear that the reviewers commented on the research “a much-needed study about the mechanisms underlying discriminatory processes in hiring by race/ethnicity and incarceration history”, on a “a timely and important topic”, and our manuscript “structures in a clear and articulate way”. 

We found the thoughtful and generous suggestions by you and the two reviewers to be very helpful as we revised our manuscript. We believe the revised manuscript is much stronger thanks to the feedback. We outline how we incorporated the suggestions and addressed the concerns below. 

Major concerns/suggestions

Reviewer 1 pointed out they’d like to see us providing more context demonstrating the (un)employment consequences of incarceration, and support for Hypothesis #1. 

We are grateful for Reviewer 1 for this general suggestion. We have revisited this part of our introduction, and have enriched the context leading up to Hypothesis #1. For instance, on Page 4, we have cited further - and recent - research (also thanks to suggestions from Reviewer #2) demonstrating unemployment consequences of incarceration history (e.g., Reich, 2007; DeWitt & Denver, 2020). Importantly, research mostly converged to demonstrate a clear employment disadvantage for individuals with incarceration history, despite the different contexts, jobs, candidate race manipulations, etc. examined in each research study cited here. 

Review 1 points out that they would like to see us elaborate on the section on intersectionality. 

In the revised manuscript, we have now added more substance (such as on Pages 9-10), while also pointing out that there has been little to none theoretical or empirical work regarding intersectionality between race and incarceration history. For this reason, we drew from the gendered race theoretical perspective as a general framework for making predictions regarding how race and incarceration history could interact to inform hiring decisions. 

Review 1 asked for more clarification and delineation of the “warmth, competence, morality, sociability” constructs. 

We appreciate Review 1 pointing out that in our original manuscript, we were using “warmth” and “sociability” interchangeably to refer to the same dimension of social perception. This is mostly because of the inconsistency of terminology in previous literature - the same dimension has been referred to by both/either terms (for instance, “warmth” in Young & Powell, 2015, and “sociability” in Brambilla et al., 2011). 

In the revised manuscript, we make it clear that the warmth/sociability dimension is one of the three dimensions of person perception. We have changed the language throughout the manuscript to be consistent and clear. 

Reviewer 2 pointed out that the justification of “intergroup threat” was thin in the original manuscript, and suggested focusing on “perceived risk for liability and safety reasons”. Here Reviewer #2 also suggested additional research.

First of all, we are grateful for Reviewer 2’s suggestions of additional literature to enrich our literature review and introduction in general. We have consulted the suggested research, and have incorporated them into our narrative. For instance, we found that incorporating Reich (2017) helped us strengthen the background discussion on the effect of incarceration history on willingness to employ, which also addresses Reviewer 1’s suggestion (explained above) of enriching the literature on this relationship (incarceration history and employment outcomes). 

In addition, we also incorporated recent work from DeWitt & Denver (2020), as suggested by Reviewer #2. Specifically, on Page 6, we have added: “Most recently, DeWitt & Denver (2020) found that while job applicants with an incarceration history were viewed more negatively in the hiring context, race (Black or White) did not moderate such an effect. Importantly, DeWitt & Denver (2020) posited several possible reasons for the null effect of race, such as their survey methodology, or the possibility that certain social categories could dominate our perception when encountering multiple intersectional social categories (DeWitt & Denver, 2020).”

Moreover, we believe that the inclusion of this work also helps add to comment #2 above (from Reviewer 1) concerning intersectionality. On Page 9, we added further clarification regarding this point “It is also possible, as DeWitt & Denver (2020) pointed out in their writing, that certain social categories could dominate our perception when encountering multiple intersectional social categories. For instance, Rattan and colleagues (2019) found that evaluation of job candidates depended on which social category was more salient in the given context, even when the job candidates were perceived to belong to multiple social categories. In a stereotypically male and stereotypically Asian employment context, participants rated an Asian American female applicant as more hirable when her race, rather than gender, was made salient (Rattan et al., 2019).”

In considering our research in light of the comments from reviewers and additional literature we’ve reviewed during revision, we’ve now eliminated the measures of perceived threat (reflected on Page 14 as well as in our hypotheses, model and power estimates). Our prior conceptualization of threat emphasized interpersonal threat, which may be less relevant for professional positions and hiring decisions made by people who may not personally interact with the job candidate. This will also allow us to better focus on the core person perception and theory we plan to test. 

In addition, we also had extensive discussion on the role of perceived risk for liability and safety reasons, and consulted the additional literature suggested by Reviewer 2. While we agree that perceived risk for liability and safety could play a role in hiring individuals with incarceration history, we eventually decided that such measures would not be the best fit in the current study. Because our participants are instructed to put themselves in the position of a hiring manager evaluating the quality and fit of the job application, and not in the position of HR personnel who are typically responsible for the legal aspects of hiring. For this reason, we don’t think perceived risk for liability and safety is of central concern in this specific context. 

Reviewer #2 asked for more justification for our name selections. 

First of all, we agree with the Reviewer’s point about using an “Americanized” first name for our Asian job candidate, so that it does not introduce the possible confound of activating the stereotype regarding English fluency. While it is unfortunate and probably unavoidable that an Asian name (even Americanized first name with an Asian last name) would activate foreigner stereotypes (and relatedly, stereotypes regarding English fluency), we agree that using an Americanized first name (e.g., Kevin) would help temper this possible confound. In addition, we appreciate the additional source that Reviewer #2 shared (Gladdis, 2017). While we agree and fully recognize that different Black names may signify different levels of socioeconomic class, we must prioritize a strong manipulation of ethnicity. Of course, as Gladdis (2017) points out, the more Black stereotypical names tend to be perceived to belong to lower SES, and we recognize that this may be a limitation of the current study. To provide further justification and clarification along these lines, on Page 14, , we added “For instance, in a previous study, the Black name Jamal was perceived as black among more than 95% of respondents (Gladdis, 2017a). In a separate study, it was shown that when Hispanic first names were combined with Hispanic last names (as opposed to Anglo last names), the individual was perceived as Hispanic more than 90% of the time (Gladdis, 2017b).”

We also now make an effort to address Reviewer 2’s comment regarding perceived SES of the names. For instance, on Page 14, we add: “However, to offset potential covariance of minoritized names with lower SES, we used an atypical White name of Kody, because atypical names are more likely to be given by mothers with lesser education (Barlow & Lahey, 2018).”

Before we conduct the study, we would pilot our chosen names to make sure that the vast majority of participants would attribute the ethnicity correctly according to our intended manipulation. Moreover, we think that social class signal is quite interesting in this context, and would love to explore this factor further in follow up work. We have now added some clarification in our manuscript, and have pointed out this limitation in the Discussion section. 

Minor concerns/suggestions:

Reviewer #1 pointed out that by manipulating incarceration through signaling the applicant as the founder and president of an organization for formerly incarcerated people, it is possible that this would make the results potentially more conservative. We agree with the reviewer’s comment in that being the founder and president of such an organization, in addition to manipulating incarceration status, could have possibly also signaled positive characteristics (especially in morality, competence, warmth, etc.). Therefore, we’ve changed this manipulation to member and Secretary of the organization. This change is reflected in the revised manuscript. 

Reviewer #1 asked for clarification about the framing of the study’s overall aims, which were previously indicated as better understanding mechanisms of the unemployment collateral consequence and its disparate impact on people of color. We revised our aim to emphasize the psychological mechanisms we’re interested in. This aim now reads “Although past research has examined main and interactive effects of criminal history and ethnicity on biased hiring outcomes, little is known about the psychological mechanisms through which these biases may occur. Therefore, the current study seeks to better understand the psychological mechanisms for the unemployment collateral consequence.”

Review #1 suggested that we use either Latinx or Chicanx throughout the manuscript for consistency and to avoid confusion. We agree with this point and now refer to this level of our ethnicity manipulation “Latinx”. 

Reviewer #1 asked about our reasoning for focusing on human resource managers. To clarify, we are recruiting individuals who are in the position to hire (e.g., hiring managers), and NOT specifically HR managers. We have added clarifications in the manuscript as well (Page 13).

Reviewer #1 also asked about our reasoning for choosing this particular job posting. Here we’d like to clarify that we had intended to use a white collar job, which adds to the literature on hireability of previously incarcerated individuals focusing primarily on blue collar jobs. We also planned to use an entry level position, in order to allow room for variability among hireability judgments. Lastly, we adapted a job ad from a similar position at our university to increase ecological validity. We added clarifications in our manuscript (Page 14).

Reviewer #2 suggested adding additional questions to Appendix M about the size of respondents’ departments/units in which they were hiring managers and the number of years they’ve been in these roles. We have added two questions to the Demographic Survey for the size of respondents’ departments/units as well as the number of years they have been in a role in which they hire other employees.

Reviewers #1 and #2 indicated a lack of description for data access. We added a paragraph at the beginning of the methods section describing pre-registration and access to data. 

We thank you and the two reviewers for the valuable suggestions that have improved the manuscript. Naturally, we have also re-read and edited the entire paper to make additional improvements, fixed some typos. streamlined the paper after making the suggested changes, and edited formatting to adhere to the publication guidelines of the journal. We believe the paper is much stronger than our initial submission and hope it is ready for publication.

Sincerely, 

Christopher Beasley, Ph.D. Y. Jenny Xiao, Ph.D.

University of Washington, Tacoma University of Washington, Tacoma

---

## [Decision Letter · Decision Letter 1]

28 Dec 2022

Incarceration History and Ethnic Bias in Hiring Perceptions: An Experimental Test of Intersectional Bias & Psychological Mechanisms

PONE-D-22-02329R1

Dear Dr. Beasley,

We’re pleased to inform you that your manuscript has been judged scientifically suitable for publication and will be formally accepted for publication once it meets all outstanding technical requirements.

Kind regards,

Andrea Knittel

Academic Editor

PLOS ONE

Additional Editor Comments (optional):

Reviewers' comments:

Reviewer's Responses to Questions

**Comments to the Author**

1. Does the manuscript provide a valid rationale for the proposed study, with clearly identified and justified research questions?

Reviewer #2: Yes

2. Is the protocol technically sound and planned in a manner that will lead to a meaningful outcome and allow testing the stated hypotheses?

Reviewer #2: Yes

3. Is the methodology feasible and described in sufficient detail to allow the work to be replicable?

Reviewer #2: Yes

4. Have the authors described where all data underlying the findings will be made available when the study is complete?

Reviewer #2: Yes

5. Is the manuscript presented in an intelligible fashion and written in standard English?

Reviewer #2: Yes

6. Review Comments to the Author

You may also provide optional suggestions and comments to authors that they might find helpful in planning their study.

Reviewer #2: The authors have adequately addressed and incorporated all feedback.

7. PLOS authors have the option to publish the peer review history of their article (what does this mean?). If published, this will include your full peer review and any attached files.

Reviewer #2: No

---

## [Editor Report · Acceptance letter]

6 Jan 2023

PONE-D-22-02329R1 

Incarceration History and Ethnic Bias in Hiring Perceptions: An Experimental Test of Intersectional Bias & Psychological Mechanisms 

Dear Dr. Beasley:

I'm pleased to inform you that your manuscript has been deemed suitable for publication in PLOS ONE. Congratulations! Your manuscript is now with our production department. 

Kind regards, 

on behalf of

Dr. Andrea Knittel 

Academic Editor

PLOS ONE